# Hospitalization and Mortality by Vaccination Status among COVID-19 Patients Aged ≥ 25 Years in Bangladesh: Results from a Multicenter Cross-Sectional Study

**DOI:** 10.3390/vaccines10121987

**Published:** 2022-11-23

**Authors:** Md. Saydur Rahman, Md. Golam Dostogir Harun, Shariful Amin Sumon, Tahrima Mohsin Mohona, Syed Abul Hassan Md Abdullah, Md. Nazuml Huda Khan, Md. Ismail Gazi, Md. Saiful Islam, Md. Mahabub Ul Anwar

**Affiliations:** 1Department of Administration, DNCC Dedicated COVID-19 Hospital, Dhaka 1212, Bangladesh; 2Infectious Diseases Division, International Centre for Diarrhoeal Disease Research, Bangladesh (ICDDR, B), Dhaka 1212, Bangladesh; 3Department of Public Health, Daffodil International University, Dhaka 1341, Bangladesh; 4SafetyNet Bangladesh, Dhaka 1212, Bangladesh; 5Department of Administration, Kurmitola General Hospital, Dhaka 1206, Bangladesh; 6Sheikh Russel National Gastroliver Institute and Hospital, Dhaka 1212, Bangladesh; 7School of Medical Science, University of New South Wales, Sydney 2052, Australia; 8Office of Health Affairs, West Virginia University, Morgantown, WV 26506, USA

**Keywords:** hospitalization, mortality, COVID-19 vaccination, World Health Organization, Bangladesh

## Abstract

The COVID-19 pandemic has inflicted a massive disease burden globally, involving 623 million confirmed cases with 6.55 million deaths, and in Bangladesh, over 2.02 million clinically confirmed cases of COVID-19, with 29,371 deaths, have been reported. Evidence showed that vaccines significantly reduced infection, severity, and mortality across a wide age range of populations. This study investigated the hospitalization and mortality by vaccination status among COVID-19 patients in Bangladesh and identified the vaccine’s effectiveness against severe outcomes in real-world settings. Between August and December 2021, we conducted this cross-sectional survey among 783 RT-PCR-confirmed COVID-19 hospitalized patients admitted to three dedicated COVID-19 hospitals in Bangladesh. The study used a semi-structured questionnaire to collect information. We reviewed the patient’s records and gathered COVID-19 immunization status from the study participants or their caregivers. Patients with incomplete or partial data from the record were excluded from enrollment. Logistic regression analyses were performed to determine the association between key variables with a patient’s vaccination status and mortality. The study revealed that overall hospitalization, severity, and morality were significantly high among unvaccinated study participants. Only one-fourth (25%) of hospitalized patients were found COVID-19 vaccinated. Morality among unvaccinated COVID-19 study participants was significantly higher (AOR: 7.17) than the vaccinated (11.17% vs. 1.53%). Severity was found to be seven times higher among unvaccinated patients. Vaccination coverage was higher in urban areas (29.8%) compared to rural parts (20.8%), and vaccine uptake was lower among female study participants (22.7%) than male (27.6%). The study highlighted the importance of COVID-19 vaccines in reducing mortality, hospitalization, and other severe consequences. We found a gap in vaccination coverage between urban and rural settings. The findings would encourage the entire population toward immunization and aid the policymakers in the ground reality so that more initiatives are taken to improve vaccination coverage among the pocket population.

## 1. Background

The COVID-19 pandemic has inflicted a massive disease burden globally [1], involving 623 million confirmed cases and 6.55 million deaths as of 4 October 2022. The severity of coronavirus disease 2019 (COVID-19) can range from mild symptoms to serious consequences, especially in the elderly and with comorbidities [2,3]. The disease is highly infectious, having prompt clinical manifestations in cases where the principal transmission route is respiratory, i.e., via droplets and contact routes [4,5,6]. Several infection prevention and control (IPC) measures, including performing regular hand hygiene, maintaining six feet of social distancing, appropriately wearing the face mask, and contact tracing when a positive case is detected, can restrict the viral spread of severe acute respiratory syndrome coronavirus 2 (SARS-CoV-2) infections [7]. However, non-clinical initiatives alone cannot limit the transmission of this deadly virus [8]. Both the World Health Organization (WHO) and the Centers for Disease Control and Prevention (CDC) strongly recommend vaccination against COVID-19 as the safest long-term prophylactic approach to containing the pandemic [9,10]. The COVID-19 vaccinations have started, which make a difference and have proven significantly effective at preventing infection with different SARS-CoV-2 variants [11,12]. Vaccines develop the body’s natural defenses and boost the immune system to combat the virus. Several vaccine trials achieved remarkable immunity for the COVID-19 vaccine, which were found to be up to 92% among those previously infected, 87% among hospitalized patients, and 92% among those with severe disease [13,14,15]. The effectiveness of these vaccines among hospitalized patients has been a topic of discussion in different forums.

Evidence showed that vaccines significantly reduced the infection, severity, and mortality across a wide age range of populations in developed and developing countries [13,16]. A recently conducted case-control study in the United States of America (USA) showed that fully vaccinated adults (≥65 years) had 94% effectiveness against COVID infections, whereas partially vaccinated patients had 64% effectiveness against COVID infections and the findings were statistically significant [17]. Another community-based study conducted in Los Angeles showed that fully vaccinated personnel were less likely to be admitted to the hospital, shifted to the intensive care unit (ICU), require mechanical ventilation, and die than partially vaccinated ones due to COVID infections [18]. A similar result was found in other countries as well, such as England and Israel, where vaccinated patients had fewer risks of progressing to a severely diseased state and requiring ventilation in hospitals than those who were non-vaccinated or partially vaccinated [19,20]. In Malaysia, the age-standardized mortality rate for unvaccinated individuals was 43.2 times higher than for individuals fully vaccinated [21]. A study in India reported that mortality was significantly less among the vaccinated population [22].

In Bangladesh, over 2.02 million clinically confirmed cases of COVID-19, with 29,371 deaths, have been reported to WHO thus far [23]. Bangladesh is one of the most densely populated countries where strict social distancing is difficult. Being at high risk of transmission of SARS-CoV-2, the vaccine may be the most effective containment strategy for the country. The Government of Bangladesh commenced the COVID-19 vaccination program on 27 January 2021, with mass immunization in the following months [24]. Acceptance of the COVID-19 vaccine among Bangladeshi inhabitants was reported to be 80%, and the government has announced a target of vaccinating three-fourths of its population by 2022 [25]. However, nearly one-quarter (22.3%) of the population still remains unvaccinated [26]. Yet, no explicit study has been conducted on hospitalization and mortality by vaccination status related to COVID-19 in Bangladesh. This study aims to investigate the hospitalization and mortality by vaccination status among COVID-19 patients in Bangladesh and to identify the vaccine’s effectiveness against severe outcomes in real-world settings. The findings of this study will contribute to the overall success of the vaccination program, tailored tactics, and vaccine promotion campaigns aimed at the general community in Bangladesh.

## 2. Methods

### 2.1. Study Setting

We conducted this survey in three COVID-dedicated public hospitals in Bangladesh: Dhaka North City Corporation (DNCC) Dedicated COVID-19 Hospital, Sheikh Russel National Gastroliver Institute and Hospital, and Kurmitola General Hospital. All three hospitals have a central oxygen supply available and, as they are referral hospitals, provide care to the most critical patients, referred from all over Bangladesh. DNCC Dedicated COVID-19 Hospital is the only specialized hospital in Bangladesh for COVID-19 patients built to combat the pandemic. It is the largest COVID hospital, with 1500 beds, including 212 ICU beds and 288 high-dependency units. Sheikh Russel National Gastroliver Institute and Hospital is a 500 bedded specialized hospital with an annual patient turnover of 5000, and Kurmitola General hospital is a 500-bedded general hospital with an annual patient turnover of 9647 in 2021, both of which were announced to be COVID-dedicated hospitals by the government hospital during the pandemic.

### 2.2. Study Design and Population

This cross-sectional study was carried out among hospitalized patients aged ≥ 25 years with SARS-CoV-2 infection confirmed by reverse transcription–polymerase chain reaction (RT-PCR) test from August to December 2021. We enrolled people aged 25 years and above because at the time the study was conducted, only this group was eligible for the COVID-19 vaccine. Patients or caregivers who were unwilling to participate in the study were excluded.

### 2.3. Sampling Technique

Patients admitted with confirmed cases of COVID-19 in the study period comprised the study population. Hospital admission records were assessed every day to form a sampling frame. A systematic random sampling method was used to enroll every third patient from the sample frame. If any participant was unwilling to participate, we took the following patient in the sampling frame.

### 2.4. Data Collection

A semi-structured questionnaire was used to obtain information. We reviewed the patient’s records from the patient file and hospital registry to extract information on COVID-19 hospitalization status as well as the outcome of patients. Both nasopharyngeal or oropharyngeal swabs were collected by a medical technologist specifically instructed and trained in the techniques of real-time PCR and in vitro diagnostic procedures for laboratory testing. National guidelines on the standard procedure for conducting RT-PCR test was followed. In addition, we collected data on patients’ symptoms during hospitalization, and comorbid conditions, including ICU and oxygen requirements.

The study team also gathered COVID-19 immunization status, number of doses, and sociodemographic information from the study participants or their caregivers. The vaccination status of both single and double doses was confirmed by the patient or their caregivers or through a vaccination card authorized and issued by the Government of Bangladesh. The patient’s discharge was determined after the results of the RT-PCR test showed negative status along with clinical recovery. Patients with incomplete or partial data from the record were excluded from enrollment.

### 2.5. Statistical Analysis

The study used descriptive statistics for continuous and categorical variables with percentages, mean, and standard deviations to illustrate sociodemographic information. We also compared the hospitalization rate and outcome, including deaths among vaccinated and non-vaccinated patients. Additionally, we categorized the severity of patients considering comorbidities, ICU, and oxygen requirements. We conducted logistic regression to determine the association between key variables, patient vaccination status, and mortality. Both adjusted and unadjusted odds ratios within a 95% confidence interval were calculated. We performed all the statistical analysis in STATA software (Stata Corp, College Station, TX, USA, 2013, version 13.0).

### 2.6. Ethical Consideration

We collected written permission from the hospital authority to collect clinical information from the patient medical records. Before collecting demographic information, we obtained written informed consent from the patients or the accompanying family caregivers. The study-maintained anonymity and confidentiality strictly. This study was reviewed and approved by the Institutional Review Board (IRB) of Dhaka North City Corporation Dedicated COVID-19 Hospital.

## 3. Results

### 3.1. Key Characteristics of Study Participants by Hospitals

We enrolled a total of 783 RT-PCR confirmed COVID-19 hospitalized patients in the study. The median age of the participants was 56.0 years (IQR: 47.0–66.0), and half of them were female. Almost half (51.1%) of the study participants were from rural areas. The study documented that only 25.0% (196/783) of hospitalized patients received the COVID-19 vaccine. Among the vaccinated participants, 63.3% (124/196) took double doses. The study revealed that 71.8% of the enrolled participants had a history of the co-morbid condition. However, the mortality among enrolled hospitalized patients was 7.9% during the study period (Table 1).

### 3.2. Key Characteristics of Study Participants by Vaccination Status

Table 2 describes the key characteristics of enrolled hospitalized patients based on vaccination status. The overall vaccination coverage was low among the study participants, whereas males had slightly higher (27.6%) vaccination than their counterparts (22.7%). Similarly, patients living in rural areas had a lower vaccination uptake compared to urban patients (20.8% vs. 29.8%). Three-fourths (73.7%) of the study participants had comorbidities. The study documented that oxygen and ICU requirements were significantly higher (76.1% and 75.4%) among unvaccinated patients. In terms of severity, considering patients had comorbidities, oxygen, and ICU requirements, the majority of the participants (73.8%) were not vaccine recipients. Unvaccinated patients had remarkably higher mortality (95.2%) compared to the COVID-19 vaccine recipients (4.8%) during the study period (Table 2).

Figure 1 shows the number of deaths among hospitalized patients by vaccination coverage during the study period. The mortality was found to be higher among COVID-19 unvaccinated patients (10.1%, 59/587) compared to the vaccinated hospitalized patients (1.53%, 3/196) (Figure 1).

### 3.3. Factors Associated with COVID-19 Mortality of Enrolled Participants

The study found a significant association between receiving the COVID-19 vaccine and a reduction in COVID-19 mortality among hospitalized patients (Table 3). Unvaccinated hospitalized patients were seven times more likely to die from COVID-19 (AOR: 7.17, 95%CI: 2.21–23.27, *p*-value: 0.001) than vaccinated, hospitalized patients. The adjusted odds of death were found to be 9.98 times higher among unvaccinated COVID patients with co-morbidities compared to vaccinated patients who had co-morbid conditions (AOR: 9.98, 95%CI: 2.39–41.68, *p*-value: 0.002). Among patients who required oxygen (O_2_), unvaccinated patients had higher odds of death (AOR: 10.24) than their counterparts (95%CI: 2.46–42.69, *p*-value: 0.001). We found that unvaccinated, hospitalized patients with ICU requirements were at higher risk for COVID-19-associated mortality (AOR: 7.74, 95%CI: 1.78–33.74, *p*-value: 0.006) than the vaccinated patients with similar requirements. Considering the severity, the odds of mortality (AOR: 6.98, 95%CI: 1.57–30.92) among unvaccinated patients were statistically significant compared to vaccinated patients (Table 3).

## 4. Discussion

This multicenter cross-sectional study evaluated the impact of the COVID-19 vaccine against severity, hospitalization, and mortality among the patients admitted into three dedicated COVID-19 hospitals in Bangladesh. This study revealed that hospitalization, severity, and morality were significantly high among the COVID-19 unvaccinated study participants. Only one-fourth of all hospitalized COVID-19 patients were found to be vaccinated, which is much lower than the global vaccine uptake rates. A study found false claims related to the estimated deaths caused by the COVID-19 vaccine were one of the highest pieces of circulating misinformation [27]. Our study demonstrates vaccination as a protective factor, as unvaccinated hospitalized patients were 7.17 times more likely to die from COVID-19-related complications. These results could be used as supporting data to design evidence-based risk communication and community engagement strategies to promote vaccine uptake.

Hospitalized COVID-19 patients who had received at least one dose of the vaccine had lower incidences of death than unvaccinated patients (Figure 1). This was also observed in a US study where receipt of Pfizer–BioNTech COVID-19 vaccine (BNT162b2 mRNA) and Moderna (mRNA-1273) was associated with a 100% reduction in hospitalizations or mortality [28,29]. In a nationwide retrospective cohort study conducted in Hungary, vaccination prevented all-cause mortality [30]. A study in a low- and middle-income countries (LMIC) setting reported that being fully vaccinated was associated with a 70% reduction in the odds of death among hospitalized COVID patients [22]. As COVID-19 vaccines induce antibodies, it boosts the immune response against the variants of SARS-CoV-2. Vaccines might also elicit the body to produce T-cells which kill COVID-19 infected cells, thus reducing the mortality attributed to COVID-19. The study findings revealed that COVID-19 vaccination was associated with a significant reduction in the severity of COVID-19. We assessed the severity as a condition where patients had co-morbidities and oxygen and ICU requirements. Our study showed that unvaccinated COVID patients with pre-existing co-morbid conditions were ten times more likely to die from COVID-19-associated complications. The results are in line with a previous systematic review which documented that the presence of co-morbidities such as hypertension, diabetes, chronic obstructive pulmonary disease (COPD), and chronic kidney disease (CKD) increased the risk of COVID-associated mortality [31]. Our study demonstrated that the requirement for oxygen and ICU support in unvaccinated patients was almost three times more than in those vaccinated prior to hospitalization. The findings correspond to an overall 94% effectiveness of vaccines against hospitalization (mainly for oxygen support) and a reduced need for mechanical ventilation [32,33]. Unvaccinated patients requiring oxygen and ICU support also had significantly greater odds (10.24 and 7.74 times, respectively) of dying. Our findings align with prior studies that reported a 98% and 91% reduction in ICU admission among COVID vaccine recipients [29,33]. This reduction in ICU and oxygen requirement might be due to the IgG antibody response produced by vaccines that fight against the SARS-CoV-2 strains, thus reducing the severity of COVID-19-associated illness.

This analysis also found that old age is associated with increased odds of mortality, as older patients are more susceptible to SARS-CoV-2 [34]. Previously, a USA study comprising a large cohort of elderly patients (n = 20,037) documented that receiving the COVID vaccine in cirrhosis patients was associated with a 100% reduction in COVID-19-related hospitalization or death [28]. Likewise, an Israeli study reported that vaccinated patients had a higher prevalence of risk factors for COVID-19 (older age, co-morbidity) but still showed a lower rate of ICU admission and mortality compared to the unvaccinated controls [32]. COVID-19 vaccine was the only protective factor among those patients. At the same time, vaccination is critical for people of all age groups, yet, the impaired immunity in elderly individuals with co-morbid conditions makes them vulnerable to the inflammation produced by SARS-CoV-2 infection [35]. As the maximum risk of progression to severe COVID-19 cases with pre-existing conditions, vaccines protect these immunocompromised individuals.

Our data revealed a disparity in vaccination coverage between urban and rural areas (29.8% vs. 20.8%). A CDC report documented a lower COVID-19 vaccination coverage in rural areas (58.5%) compared to urban counties (75.4%) across the USA [36]. The government of Bangladesh launched a mass vaccination program to vaccinate 80% (over 130 million) of the country’s total population with vaccines in four different phases [37]. Although more than one hundred ten million people received the COVID-19 vaccine as of 19th September 2022 in Bangladesh, a low turnout has been recorded in rural places as a substantial gap exists concerning receiving vaccines between the two areas of people [37]. The poor IPC situation and overcrowded conditions in hospital settings might create hesitancy among people about taking COVID-19 vaccines, as the vaccination program is mostly carried out in those hospitals [38,39]. Taking the interest of the rural population into account, as approximately 85% were willing to be vaccinated, with a fraction even willing to pay for the vaccine [37], the Government of Bangladesh needs to take more initiative to increase vaccination coverage in those areas. Ensuring higher vaccination coverage is of particular importance as the majority of the population lives in rural areas.

Amidst the unrest caused by SARS-CoV-2, researchers have been trying to find a solution to put this global health menace to a stop [40]. Additionally, as a clear majority of people across developing countries still remain vulnerable to COVID-19 [41], vaccination seems to be the only effective clinical approach to reducing the mounting burden of this outbreak. Vaccines not only work as impactful barriers against the spread of preventable infectious diseases, but also assure routine healthcare service provision [42,43] and, therefore, should be vehemently promoted.

We conducted this survey in the three dedicated COVID-19 hospitals. We collected data from patients’ records and hospital registries, so recall bias is not an issue. However, we could not collect data on individual co-morbidities and types of COVID-19 vaccine, as this information was not available in their medical files. All the study hospitals were from Dhaka, so lack of generalization could be an issue. This is also reflected in the large confidence interval, which suggests that the sample may not provide an adequate representation of the population mean. Secondly, due to the cross-sectional study design, causation could not be measured.

## 5. Conclusions

Our study investigated the effectiveness of the COVID-19 vaccine against severe associated outcomes. Our findings highlighted the importance of COVID-19 vaccines in reducing mortality, hospitalization, and other severe consequences. Our results also highlighted the gap in vaccination coverage between urban and rural settings. The findings would inform the policymakers about the ground reality so that more initiatives are taken to improve vaccination coverage among the pocket population.

## Figures and Tables

**Figure 1 vaccines-10-01987-f001:**
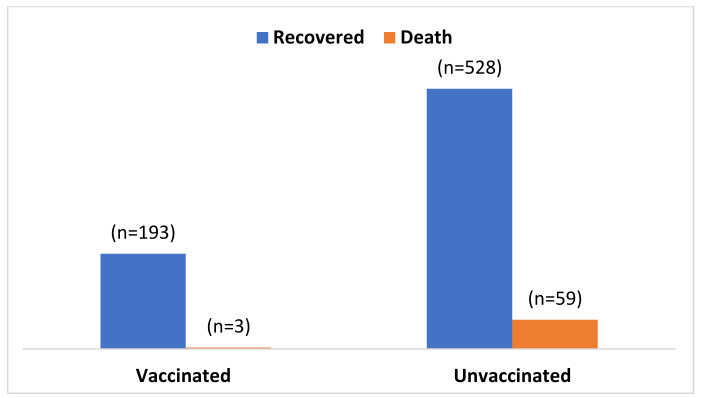
Morality by COVID-19 vaccination status among hospitalized patients.

**Table 1 vaccines-10-01987-t001:** Key characteristics of study participants by hospital.

Characteristics	Total [N = 783]	Hospital 1 [N = 344]	Hospital 2 [N = 322]	Hospital 3 [N = 117]
	*n*/N (%)			
Age (in years)				
Median (IQR)	56.0 [47.0–66.0]	58.5 [50.0–58.5]	54.0 [44.0–66.0]	60.0 [50.0–68.0]
25–40	109 (13.9)	47 (13.7)	57 (17.7)	5 (4.3)
41–50	157 (20.0)	69 (20.1)	58 (18.0)	30 (25.6)
51–60	208 (26.6)	82 (23.8)	90 (27.9)	36 (30.8)
61–70	173 (22.1)	92 (26.7)	55 (17.1)	26 (22.2)
>70	136 (17.4)	54 (15.7)	62 (19.3)	20 (17.1)
Sex: Male	391 (50.0)	179 (52.0)	147 (45.6)	65 (55.6)
Religion: Muslim	726 (92.7)	321 (93.3)	296 (91.9)	109 (93.2)
Profession				
Day labor	24 (3.1)	15 (4.4)	3 (1.0)	6 (5.1)
Business	170 (21.7)	82 (23.8)	34 (10.5)	54 (46.1)
Service	153 (19.5)	84 (24.4)	38 (11.8)	31 (26.5)
Housewife	291 (37.2)	118 (34.3)	169 (52.5)	4 (3.4)
Agriculture	40 (5.1)	9 (2.6)	12 (3.7)	19 (16.2)
No employment	105 (13.4)	30 (8.7)	66 (20.5)	3 (2.6)
Living place				
Rural	400 (51.1)	197 (57.3)	148 (46.0)	55 (47.0)
Urban	383 (48.9)	147 (42.7)	174 (54.0)	62 (53.0)
Had comorbidities	652 (71.8)	216 (62.8)	256 (79.5)	90 (76.9)
Hospitalization status, by vaccine				
Unvaccinated	587 (75.0)	255 (74.1)	301 (93.5)	31 (26.5)
Vaccinated (any dose)	196 (25.0)	89 (25.9)	21 (6.5)	86 (73.5)
COVID-19 Vaccination status				
Double doses	124 (15.8)	37 (10.8)	19 (5.9)	68 (58.1)
Single doses	72 (9.2)	52 (15.1)	2 (0.6)	18 (15.4)
Final Outcome				
Recovered and discharged	721 (92.1)	313 (91.0)	294 (91.3)	112 (95.7)
Died	62 (7.9)	29 (8.4)	28 (8.7)	5 (4.3)

**Table 2 vaccines-10-01987-t002:** Key Characteristics of study participants by vaccination status.

	Hospitalized Patients, by Vaccine		
	Total (N)	Unvaccinated n (%)	Vaccinated n (%)
Gender			
Male	391	283 (72.4)	108 (27.6)
Female	392	303 (77.3)	89 (22.7)
Living area			
Urban	383	269 (70.2)	114 (29.8)
Rural	400	317 (79.2)	83 (20.8)
Had comorbidities	562	414 (73.7)	148 (26.3)
O_2_ required	634	477 (75.2)	157 (24.8)
ICU required	244	184 (75.4)	60 (24.6)
Severity	187	138 (73.8)	49 (26.2)
Died	62	59 (95.2)	3 (4.8)

**Table 3 vaccines-10-01987-t003:** Associated factors with mortality of enrolled participants.

	UOR (95%CI)	*p*-Value	AOR (95%CI)	*p*-Value
Age				
25–40	ref		ref	
41–50	1.29 (0.46–3.61)	0.623	1.61 (0.55–4.73)	0.388
51–60	0.69 (0.23–2.03)	0.497	0.67 (0.22–2.11)	0.512
61–70	1.63 (0.61–4.34)	0.328	1.44 (0.51–4.02)	0.488
>70	3.31 (1.29–8.49)	0.013	3.16 (1.16–8.58)	0.024
Hospitalization by vaccine				
Vaccinated	ref		ref	
Unvaccinated	7.19 (2.28–23.20)	0.001	7.17 (2.21–23.27)	0.001
Had Comorbidities				
Vaccinated	ref			
Unvaccinated	10.72 (2.57–44.55)	0.001	9.98 (2.39–41.68)	0.002
O_2_ required				
Vaccinated	ref		ref	
Unvaccinated	10.31 (2.49–42.75)	0.001	10.25 (2.46–42.69)	0.001
ICU required				
Vaccinated	ref		ref	
Unvaccinated	6.81 (1.59–29.24)	0.010	7.74 (1.78–33.74)	0.006
Severity				
Vaccinated	ref		ref	
Unvaccinated	6.53 (1.50–28.4)	0.012	6.98 (1.57–30.92)	0.011

UOR = unadjusted odds ratio, AOR = adjusted odds ratio; (only variables with *p* values < 0.25 on univariate were put into multivariate regression model).

## Data Availability

The authors are responsible for the data described in this manuscript. The datasets generated and analyzed are available from the corresponding author upon request. Data Can also be available from the DNCC hospital authority at shahinbd10@yahoo.com upon request.

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
