# Peer review of "Hospitalization and Mortality by Vaccination Status among COVID-19 Patients Aged ≥ 25 Years in Bangladesh: Results from a Multicenter Cross-Sectional Study"

_vaccines, 2022, doi:10.3390/vaccines10121987_

Round 1

Reviewer 1 Report

Thank you for your submission. I have some suggestions for your manuscript.

Please define abbreviations before using.  Page 2 line 77 should read COVID-19 not COVDI-19. line 96 has an underscore change to COVID-19. Please review for grammar.

For data collection: did you track which medications the patients were treated with? If so, that would be good to state.  Interesting a negative PCR test was required for discharge when those that test positive can have a positive PCR for up to 90 days, what was the reasoning for a negative PCR versus a negative antigen test? Did you track which vaccines were administered? Astra Zeneca, Pfizer, Moderna? that would be of interest to the reader. Did you track which co-morbidities your subjects had? It would be good the stratify that data as some co-morbidities put individuals at higher risk. Add to discussion piece as well.

Table 3 please define UOR and AOR may be have an * at bottom of Table to define abbreviations

Page 7 line 250 is missing a % sign

Limitations may want to address the large Confidence intervals - could be due to the sample size.

All references need to be corrected- missing DOI numbers, missing websites and dates accessed, some references are incomplete- Please review all your references and correct them to the proper AMA format. Correct alignment of references is needed. 

Author Response

Thank you for your submission. I have some suggestions for your manuscript.

Please define abbreviations before using.  Page 2 line 77 should read COVID-19 not COVDI-19. line 96 has an underscore change to COVID-19. Please review for grammar.

Response: Thank you for your keen observations. Abbreviations of COVID-19, SARS-CoV-2, USA, ICU, LMIC, COPD, CKD have been defined at instances of first use. The typing errors in lines 77 and 96 have been corrected.

For data collection: did you track which medications the patients were treated with? If so, that would be good to state.  Interesting a negative PCR test was required for discharge when those that test positive can have a positive PCR for up to 90 days, what was the reasoning for a negative PCR versus a negative antigen test? Did you track which vaccines were administered? Astra Zeneca, Pfizer, Moderna? that would be of interest to the reader. Did you track which co-morbidities your subjects had? It would be good the stratify that data as some co-morbidities put individuals at higher risk. Add to discussion piece as well.

Response: Thank you for your valuable input. Unfortunately, the medication, type of vaccine, and type of comorbidity were not tracked in our study. And the requirement of a negative PCR test was imposed by study hospital authorities, as per the government guidelines at the time. The study was not involved in the decision‐making process for patient discharge criteria. 

Table 3 please define UOR and AOR may be have an * at bottom of Table to define abbreviations

Response: Thank you for your kind observation. We have included at bottom of the table as advised.

Page 7 line 250 is missing a % sign

Response: Thanks, % sign has been added.

Limitations may want to address the large Confidence intervals - could be due to the sample size.

Response: Appreciate the valuable feedback, the issue of a large confidence interval has been involved in the recommendation along with justification.

All references need to be corrected- missing DOI numbers, missing websites and dates accessed, some references are incomplete- Please review all your references and correct them to the proper AMA format. Correct alignment of references is needed. 

Response: Thank you, we have revised the reference system accordingly with doi numbers. 

Reviewer 2 Report

Rahman et al investigated the hospitalization and mortality by vaccination status among COVID-19 patients in Bangladesh. Such analysis is important and needed to inform policymakers about the ground reality and to improve vaccination coverage among the pocket population. However, the manuscript should be improved. I have the following comments:

i) Please provide data about vaccines (Type of vaccine, number of vaccination) and analyze the correlation between vaccine type and frequency with the incidence of the diseases/ and severity. 

ii) the methodology of RT-PCR is not mentioned in the manuscript? 

iii) Are there data about the type of variants in the case of COVID-19 patients who are got vaccinated once or twice? 

iv) This Pandemic is called COVI-19 not CoV-2, not Covid-19, not CONDI-19, not COVID_19, please revise throughout the manuscript.

v) Line 56: please expand the full name of SARS-CoV-2, Table 1: what is the purpose to mention "Religion" in the table? Table 2:“ Had Comorbidity“  should be „Had comorbidity“.

vi) The manuscript needs to be revised for the English language.

Author Response

Rahman et al investigated the hospitalization and mortality by vaccination status among COVID-19 patients in Bangladesh. Such analysis is important and needed to inform policymakers about the ground reality and to improve vaccination coverage among the pocket population. However, the manuscript should be improved. I have the following comments:

  1. i) Please provide data about vaccines (Type of vaccine, number of vaccination) and analyze the correlation between vaccine type and frequency with the incidence of the diseases/ and severity. 

Response: This is a good suggestion and it would have been great to provide details data on vaccination status but unfortunately, we did not collect that information.

  1. ii) the methodology of RT-PCR is not mentioned in the manuscript? 

Response: Thank you for the feedback. We have added methodology for RT-PCR under data collection in methods section. 

iii) Are there data about the type of variants in the case of COVID-19 patients who are got vaccinated once or twice? 

Response: Unfortunately, there was no data collected about the type of COVID-19 variants.

  1. iv) This Pandemic is called COVI-19 not CoV-2, not Covid-19, not CONDI-19, not COVID_19, please revise throughout the manuscript.

Response: Thank you for your keen observation, COVID-19 term has been revised and made uniform throughout the manuscript.

  1. v) Line 56: please expand the full name of SARS-CoV-2, Table 1: what is the purpose to mention "Religion" in the table? Table 2: “ Had Comorbidity“  should be „Had comorbidity“.

Response: Thanks, full name of SARS-CoV-2 has been added. Changes in Table 2 has been as advised.   

  1. vi) The manuscript needs to be revised for the English language.

Round 2

Reviewer 1 Report

Thank you for your revised manuscript. It is unfortunate you did not list the co-morbidities of the subjects, it would have made it a stronger paper since you refer to co-morbidities are linked to mortality and more severe disease. Please make a statement that you did not document individual co-morbidities of subjects.  It is implied in the manuscript that you assessed them which you have not.  The reader will be looking for the data stratified when you do not have that data to do so.  

Thank you!

Author Response

Thank you so much for your kind review and valuable comments. It would have been great to document individual co-morbidities of subjects, but we could not collect that information as it was not available in their medical records. We have included this statement in the limitation section of the revised manuscript.

Reviewer 2 Report

Many thanks to the authors for revision.  Although, a lot of data are not available, I recommend publication of this manuscript. 

Author Response

Response: 

Thank you so much for your kind review and valuable comments. It would have been great to include data on individual co-morbidities and the type of vaccine the study participants took. We could not collect that information as it was not available in their medical records. We have included this statement in the limitation section of the revised manuscript.

Comments: English language and style are fine/minor spell check required:

Response: We have rechecked the language and made the necessary correction
